# Mixed Milk Feeding: A New Approach to Describe Feeding Patterns in the First Year of Life Based on Individual Participant Data from Two Randomised Controlled Trials

**DOI:** 10.3390/nu14112190

**Published:** 2022-05-24

**Authors:** Nikolaos G. Papadopoulos, Theodor A. Balan, Liandre F. van der Merwe, Wei Wei Pang, Louise J. Michaelis, Lynette P. Shek, Yvan Vandenplas, Oon Hoe Teoh, Alessandro G. Fiocchi, Yap Seng Chong

**Affiliations:** 1Division of Infection, Immunity and Respiratory Medicine, The University of Manchester, Manchester M13 9PL, UK; ngpallergy@gmail.com; 2Allergy Department, 2nd Paediatric Clinic, National and Kapodistrian University of Athens, 11527 Athens, Greece; 3Danone Nutricia Research, 3584 CT Utrecht, The Netherlands; theodor.balan@danone.com; 4Department of Obstetrics and Gynaecology, Yong Loo Lin School of Medicine, National University of Singapore, National University Health System, Singapore 119228, Singapore; obgpww@nus.edu.sg (W.W.P.); obgcys@nus.edu.sg (Y.S.C.); 5Department of Paediatric Immunology, Allergy, and Infectious Diseases, Great North Children’s Hospital, Newcastle upon Tyne Hospitals NHS Foundation Trust, Newcastle upon Tyne N1 4LP, UK; louise.michaelis@nhs.net; 6Population Health Sciences Institute, Faculty of Medical Sciences, Newcastle University, Newcastle upon Tyne NE2 4HH, UK; 7Singapore Institute for Clinical Sciences, Agency for Science, Technology and Research, Singapore 117609, Singapore; paeshekl@nus.edu.sg; 8Department of Paediatrics, Yoo Loo Lin School of Medicine, The National University of Singapore and National University Health System, Singapore 119228, Singapore; 9Vrije Universiteit Brussel, UZ Brussel, KidZ Health Castle, 1090 Brussels, Belgium; yvan.vandenplas@uzbrussel.be; 10Department of Paediatrics, KK Women’s and Children’s Hospital, Singapore 229899, Singapore; teoh.oon.hoe@singhealth.com.sg; 11Translational Research in Pediatric Specialities Area, Division of Allergy, Bambino Gesù Children’s Hospital, Istituto di Ricovero e Cura a Carattere Scientifico, Piazza Sant’Onofrio 4, 00165 Rome, Italy; alessandro.fiocchi@allegriallergia.net

**Keywords:** breastfeeding, formula feeding, infant, k-means clustering, mixed milk feeding

## Abstract

‘Mixed Milk Feeding’ (MMF), whereby infants are fed with both breastmilk and infant formula during the same period, is a common feeding practice. Despite its high prevalence, knowledge regarding MMF practices and their association with (health) outcomes is limited, potentially because MMF behaviours are highly variable and difficult to standardise longitudinally. In this paper, we applied a statistical clustering algorithm on individual infant feeding data collected over the first year of life from two clinical trials: ‘TEMPO’ (*n* = 855) and ‘Venus’ (*n* = 539); these studies were conducted in different years and world regions. In TEMPO, more than half of infants were MMF. Four distinct MMF clusters were identified: early exclusive formula feeding (32%), later exclusive formula feeding (25%), long-term MMF (21%), and mostly breastfeeding (22%). The same method applied to ‘Venus’ resulted in comparable clusters, building trust in the robustness of the cluster approach. These results demonstrate that distinct MMF patterns can be identified, which may be applicable to diverse populations. These insights could support the design of future research studying the impact of infant feeding patterns on health outcomes. To standardise this in future research, it is important to establish a unified definition of MMF.

## 1. Introduction

The mode of feeding. whereby a combination of breastfeeding (BF) and formula feeding (FF) is provided to an infant or young child during the same period, is a common and widespread feeding practice. Its global prevalence during the first year of life has been estimated to be between 23 and 32%, with the highest prevalence in infants aged 4–6 months [1]. To date, there is no shared and agreed international definition or naming convention for feeding a combination of BF and FF. The World Health Organization restricts mixed milk feeding to infants aged 0–5 months and includes formula and/or animal milk (e.g., cow’s milk, goat’s milk, evaporated milk, or reconstituted powdered milk) in its definition [2]. Thulier combines FF with animal milks and food in her classification [3], and yet others refer to “combination feeding” as “daily BF and FF begun in the first week of life” [4]. As an operational, working definition, we used the previously described term, mixed milk feeding (MMF), throughout to refer to a combination of BF and FF given to a term infant during the same period [1].

Despite the substantial proportion of infants receiving MMF, knowledge regarding its association with short- and long-term health outcomes, for both mother and child, is sparse. Most research and systematic reviews on infant milk feeding has centred around exclusive forms of BF and FF, or “any BF”. Data on MMF are often lacking or unstructured. Neves et al.’s 2021 publication on the global rates and trends in the consumption of breast milk, formula, and animal milk in 113 countries, for instance, reports on previously collected feeding rates data for exclusive BF, any BF, and FF and animal milk feeding [5]. Data on the rates of combinations of feeding methods or on MMF are not included. However, between 2000 and 2018, both BF and FF rates appeared to increase in the ~25 upper-middle-income countries included in the publication. An increase in MMF rates in these countries may explain part of this dual increase. In another article, Neves et al. describe concomitant increases in exclusive BF and formula consumption under 6 months, and in BF and formula consumption at 1 and 2 years with increased maternal education in low- and middle-income countries in Eastern Europe, Central Asia, Latin America, and the Caribbean [6]. The authors state that these concomitant increases may, to some extent, reflect an increase in MMF practices in these countries. Another study conducted in the UK showed that BF initiation rates have increased over the past two decades, but the number of mothers who exclusively BF their child has failed to rise. A Korean study did, as an exception, measure MMF rates and found clear increases between 2000 and 2012 [7]. Due to the fact of its widespread and, in some countries, apparent increasing practice, MMF and its effects on health and other outcomes is gaining interest and relevance.

It is well known that BF offers several well-described health benefits to both infants and mothers and the World Health Organization therefore recommends exclusive BF until the age of 6 months [8]. BF reduces child infections and promotes healthy infant growth, amongst others [9,10,11,12,13,14,15,16,17]. Maternal benefits of nursing include decreased risk of breast cancer and type 2 diabetes [17,18,19]. BF has also been associated with improved mother–child interaction such as heightened maternal sensitivity and reduced maternal perceived stress and negative mood [20,21]. Due to the short- and long-term benefits of exclusive BF for both mother and child, continued effort should be invested to support exclusive BF during the first 6 months of life. At the same time, as MMF is commonly practiced yet poorly understood, it remains relevant to understanding how BF benefits are affected once MMF is introduced. In many of the abovementioned studies and other research informing BF recommendations [22], the benefits that exclusive vs. nonexclusive BF brings have been well documented. However, in these studies, nonexclusive BF, or mixed BF, is often treated as a homogenous group. The mixed BF group generally includes infants given BF in combination with complementary foods and liquids, including juices, formula, other milks, other liquids, or solid foods [22], leaving unclear what the contribution of only MMF specifically was in this comparison. Further to understanding if and how BF benefits are affected once MMF is introduced, it is relevant to understand whether these benefits depend on how it is done (e.g., at which age MMF is introduced and in what volumes). For instance, the European Academy of Allergy and Clinical Immunology (EAACI) Task Force suggests avoiding supplementing with (cow’s milk) formula in the first week after birth to prevent cow’s milk allergy [23].

The UNICEF UK Guide to the Baby Friendly Initiative Standards emphasises that it is necessary to support mothers to make informed decisions regarding the introduction of food or fluids other than breast milk. Mothers who BF should be provided with information regarding why exclusive BF leads to the best outcomes for their baby, but also that when exclusive BF is not possible, continued BF is important, even if partial [24]. In addition, the question remains in how far MMF—particularly MMF according to particular volumes, timing, or practices—can bring benefits over exclusive FF, or even be instrumental in supporting continued BF or a return to exclusive BF. For instance, early limited formula supplementation has been shown to improve the incidence of exclusive BF at 3 months in infants at risk of BF problems [25].

MMF, however, due to the fact of its heterogeneity in real-life settings, is neither easy to structurally classify nor analyse. In practice, mothers may adapt their MMF patterns from day to day or month to month, and different mothers follow different practices, making the study of MMF effects on health and development far from straightforward. MMF encompasses a plethora of variability that ranges from one end of a spectrum to another with regards to the proportions of BF and FF given; timing of introduction of FF; duration of BF, MMF, and exclusive FF; the composition of the formula given; how these factors evolve over time. Moreover, due to the differing classifications and definitions of MMF, comparisons across studies are often impossible.

Two previously conducted infant formula intervention studies had a design that facilitated continued BF during the study, whether or not alongside FF. The current study aimed to evaluate whether distinct and clearly defined MMF patterns could be identified in these two trials. Using feeding data from the first study (‘TEMPO’)—a recent allergy prevention randomised controlled trial conducted predominantly in Caucasian infants aged 0–12 months, we followed a statistical multivariate clustering approach to structure MMF data and identify groups of subjects with similar observed feeding patterns over time. We validated the robustness of the outcomes of the applied statistical method by reproducing the results in the second study (‘Venus’): a randomised controlled trial carried out in a population of healthy Asian infants [26].

## 2. Materials and Methods

### 2.1. Tempo Study Description

TEMPO is a prospective, randomised, controlled, multicentre study that was conducted in 13 different countries in Europe and Asia. The study was registered in the clinicaltrials.gov registry with identifier: NCT03067714 on 1 March 2017. The study population comprised 855 healthy term infants at high risk of developing allergy, who were enrolled before 112 days of age (i.e., 16 weeks). Subjects who started FF before the age of 16 weeks were randomised to one of the two intervention arms: test (partially hydrolysed whey-protein-based infant formula enriched with a specific mixture of prebiotic short-chain galacto-oligosacharides (scGOS) and long-chain fructo-oligosacharides (lcFOS; 9:1) and Bifidobacterium breve M-16V) and control (infant formula based on intact cow’s milk proteins). BF was allowed in parallel to FF. Subjects who had not started FF before 16 weeks were also included in a “BF reference” group. The study product was provided free of charge to the participants who decided to start FF. More details are provided in the Appendix A.

Data on feeding behaviour were collected via an electronic diary filled in weekly and questionnaires during in-person visits. Five outpatient clinic visits were scheduled, with the first one at screening (between birth and 16 weeks of age) and the last one at 52 weeks of age. Using this collected diary data, the ages at starting and stopping BF, study product use, infant formula consumed other than study product, and introduction of complementary feeding were deduced for each individual. The average daily volume of study product (‘SP’) intake was also recorded on a weekly basis, together with other data such as daily number of study product feedings and daily number of breast milk feedings. However, a breast milk meal is not a standard quantity, and the volume intake of breast milk (or of formula other than study product) were not recorded. Note that infant formula or follow-up formula were not considered complementary foods.

A day of MMF was defined as the recorded consumption of breast milk and infant milk formula on the same day, regardless of the proportion of the two, the type of infant milk formula, or of complementary feeding.

To identify a subset of MMF subjects using a straight-forward deterministic rule based on the available data, the MMF subset was defined as those subjects that had 21 or more days of recorded MMF days between birth and 1 year of life. The choice of the number of days (21) was intended to exclude subjects that may have had MMF only for a short period of time while not being highly restrictive in the context of the existing data. Subjects with less than 300 days of known BF and FF status were excluded from the analysis.

Subjects who had no or limited infant milk formula intake recorded during the first year (on ≤5% of total recorded days) were labelled as “Ref–BF”. Subjects who had no or limited breast milk intake recorded during the first year (on ≤5% of total recorded days) were labelled as “Ref–FF”. Subjects that were not part of the MMF subset or the two reference groups were labelled as “short transition”, as they typically showed transitions between BF and exclusive FF shorter than 3 weeks. This is illustrated in the Appendix A. The resulting split is shown in Table 1.

A multivariate K-means clustering algorithm [27] was applied to assign the subjects in the MMF subset in a data-driven way into 4 distinct groups, referred to as MMF clusters. The number of clusters was decided to ensure sufficiently large groups that could be clearly distinguished. The computations were carried out in R using the kml3d package [28].

Clustering algorithms require that a “distance” (or similarity) metric between two subjects is defined. Therefore, the multivariate Euclidean distance between subjects’ longitudinal feeding pattern was employed. A subject’s longitudinal feeding pattern was represented by the daily feeding status (comprising daily 0/1 binary variables for BF, FF, and MMF) and the daily estimated study product intake. In this way, the distance metric gives weight to the timing of starting and stopping FF (and MMF) as well as the study product volume intake.

The longitudinal feeding patterns were deduced from the collected data between birth and 350 days of age as follows:The daily feeding status variables were deduced from the start–stop ages for BF and FF. If for a given day a subject figured as neither FF nor BF (and before the start of complementary feeding), then the daily feeding status variables were set to “missing”. This was the case for approximately 2.4% of the data;The estimated daily study product volume intake was determined either by having it set to 0 for the days where subjects were not FF or by the average daily intake for all the days that corresponded to an e-diary entry. When the study product intake was not reported for an interval between time intervals with known study product intake, the average daily volume was filled in by interpolating between the closes known values. This was the case for 12% of the data;If study product volume intake was missing completely before a given age, first observation carried backward imputation was used. If the study product volume intake was missing completely after a given age, the last observation carried forward was used. This was the case for 8% of the data.

The reason for limiting these variables to 350 days was to avoid extrapolating after the 52 week visit for subjects who had the last visit slightly earlier than planned. The resulting clusters were described quantitatively and qualitatively, comparing the complete feeding journeys of subjects among the clusters. The distribution of a subset of baseline variables of interest was analysed, comparing the overall MMF subset to the reference groups and the MMF clusters between themselves.

### 2.2. Venus Study Description

A total of 539 subjects of Chinese, Malay, or Indian ethnicity were enrolled in Singapore (with 520 subjects either randomised or included in the BF reference group). The study recruited healthy term infants younger than 28 days of age. Subjects that introduced formula were randomised to a test product (infant formula/follow-on formula with scGOS and scFOS and Nuturis^®^) or one of two control products (infant formula/follow-on formula with scGOS and scFOS or infant formula/follow-on formula without scGOS and scFOS). The study product was provided free of charge to the participants whenever parents decided to start FF. The study was registered in the clinicaltrials.gov registry with identifier: NCT01609634. Details of the study description can be found in Shek et al. [26].

The feeding data collected in Venus were similar to that collected in TEMPO. One difference, however, is that feeding data were collected via a 7 day diary filled in by the parents before each visit. Seven visits were scheduled, from the first at 1 month to the last at 52 weeks. Due to the nature of the data collection mechanism, the estimated daily study product intake had to be filled in by interpolating from the closest known values in 53% of the data. For 18% of the data, the study product volume intake was not recorded either at the beginning of follow up or at the end of follow up, and in these cases, last observation carried forward and first observation carried backward was employed.

Other than applying the same methodology, no information was carried over from the TEMPO clustering results. The subjects were split into groups according to their observed feeding data as described in Table 2.

## 3. Results

### 3.1. MMF Clusters in the TEMPO Study

The four MMF identified clusters were relatively balanced size wise (Table 1, Figure 1). Unlike the reference groups, the MMF subjects had at least 21 days of overlapping BF and FF. After examining the corresponding feeding behaviours (as shown in Figure 2 and Figure 3), an interpretation is provided as to how these clusters related to different feeding patterns:Cluster 1: Early exclusive FF (early transition to exclusive FF) are subjects that have an early episode of mixed milk feeding, followed by exclusive FF after approximately 80 days of age. Typically, by the time of introducing complementary foods, the babies are not BF anymore. In terms of formula intake, the quantity is very similar to the subjects in Ref–FF;Cluster 2: Later exclusive FF (later transition to exclusive FF) represents a feeding pattern where a mixed milk feeding episode is usually observed as a starting before 150 days of age (and, overall, later compared to early exclusive FF), with stopping of BF close to the age of introducing complementary feeding;Cluster 3: Long-term MMF are MMF subjects who are characterised by the introduction of formula quite early and continuing both FF and BF until 1 year of age;Cluster 4: Mostly BF are MMF subjects that have a prolonged period of exclusive BF, usually later introduction of formula, and in smaller amounts. The daily number of BF meals is close to that in the Ref–BF group.

The early exclusive FF and later exclusive FF clusters correspond to feeding patterns that contain a transition towards full FF within the first 5–6 months of age. Long-term MMF and mostly BF are both patterns that include long-term BF up to 1 year of age. However, it is of interest to note that in contrast to later exclusive FF and mostly BF, both early exclusive FF and long-term MMF do have an early introduction of formula (within the first 3 months of life). For reference, the feeding patterns corresponding to Ref–BF and Ref–FF groups were also visualized in the Appendix A.

The introduction of complementary feeding (Figure 3, top left of graph) occurs slightly earlier in the formula-intensive clusters (median age of 140 in early exclusive FF, 145 in Ref–FF, and 148 in the later exclusive FF compared to 159 days in mostly BF and 167 days in Ref–BF). This is detailed in the Appendix A.

### 3.2. Replications of Findings in an Asian Population—Mixed Milk Feeding Patterns in the Venus Study

Despite the different settings of the two trials and the independent application of the clustering algorithm in Venus, the MMF clusters showed a clear correspondence with the clusters identified in TEMPO (judging by the similarities between the results in Figure 4 and those in Figure 2, and between Figure 3 and Figure 5) albeit in different proportions (Table 3).

Most MMF subjects in both trials were in the early exclusive FF cluster, showing an early introduction of FF and subsequent stopping of BF, although this was proportionally larger than in TEMPO. The later exclusive FF cluster described subjects that switched to exclusive FF around the time of introduction of complementary feeding. The long-term MMF and the mostly BF clusters were present in Venus as well, with the latter showing a slightly larger number of subjects with a very early MMF episode followed by a return to exclusive BF (and often, by a later MMF episode as well).

A few differences in observed feeding behaviour between the two populations were:In Venus, subjects in the early exclusive FF cluster had a later median age of introducing complementary feeding, a slightly higher average number of study product meals, and a higher study product intake;In Venus, the subjects in the later exclusive FF cluster introduced FF earlier and stopped BF later compared to TEMPO;In Venus, more subjects from the mostly BF cluster had some FF in the first 100 days compared to TEMPO. The mixed feeding episodes in this cluster tended to start later in Venus (at approximately 150–200 days of age) compared to TEMPO (at approximately 100 days of age)

More details can be found in the Appendix A.

### 3.3. Baseline Variables and Their Association with MMF Clusters

In both the TEMPO and Venus studies, a large number of variables were recorded at baseline (118 in TEMPO, 108 in Venus). The baseline variables were, in themselves, often correlated. For example, we found that, in both studies, paternal education level was very highly correlated with maternal education level. In Venus, where family income was recorded, it was also highly correlated with the education level. In both studies, less smoking was observed in the families with higher education levels.

With these considerations, a subset of 11 variables representative of the purposes of this paper was identified, comprising birth weight, sex, mother’s highest level of education completed, maternal weight pre-pregnancy, maternal BMI pre-pregnancy, maternal age, race, mode of delivery, number of biological siblings, maternal “any allergy”, and both parents “any allergy”. The distribution of these variables, described in the Appendix A, was compared between the MMF subset and the Ref–BF and Ref–FF groups first, and between the MMF clusters within the MMF subset second (Appendix A for TEMPO, Appendix A for Venus).

In terms of race, the two studies were markedly different. In TEMPO, the MMF subset has a lower proportion of Caucasian subjects compared to the reference groups, the clearest example being the long-term MMF cluster (Figure 6, left). In Venus, the clusters with more BF had a higher proportion of Chinese subjects (Figure 6, right).

For birth weight, maternal weight, and maternal BMI pre-pregnancy, there did not appear to be large differences among clusters. However, the average birth weight was, overall, lower for the subjects from Venus, likely due to the substantially larger proportion of Asian subjects, who are known to be lighter at birth than Caucasian infants [29,30,31]. Ref–FF was the group with the highest maternal BMI pre-pregnancy in both TEMPO and Venus, while the birth weight was highest in the Ref–BF reference group. MMF mothers tended to be older than mothers in the reference groups in Venus, while long-term MMF mothers were older compared to the MMF average in TEMPO (see Figure 7).

Maternal education was associated with the MMF clusters in the data from both studies. In both TEMPO and Venus, Ref–FF was the group with the lowest proportion of mothers with a university degree. The cluster with the largest proportion of highly educated mothers was long-term MMF in TEMPO and Ref–BF in Venus (see Figure 8).

## 4. Discussion

To our knowledge, this is the first in-depth study of MMF practices in the first year of life in which MMF behaviours are universally and structurally described. In two distinct populations of infants from different cultures and regions, similar MMF clusters and patterns could be distinguished, namely, “early exclusive FF”, “later exclusive FF”, “long-term MMF”, and “mostly BF”. This supports the robustness of the results for the purpose of describing and structuring MMF practices data.

MMF was remarkably commonly practised, with more than half of all infants studied having MMF (for at least 21 days, and often much longer). The choice of 21 days as a cut-off for defining a subject as “MMF” is semi-arbitrary. As a sensitivity analysis, other cut-offs were also employed using 0, 7, 14, and 28 days, and they resulted in clusters that generally followed the same interpretation as with the 21 day definition. Note that in the special case of 0 days, all subjects were assigned to a cluster in which case the clusters did not refer to the whole population rather than the MMF subset in the two studies.

A surprising observation was that a substantial proportion of MMF infants (~20%) received MMF for close to 12 months (“long-term MMF” cluster), indicating that they received sustained partial BF across the first year and possibly even beyond. Another ~20% of MMF infants fell in the “mostly BF” cluster, most of whom were also still being partially BF at the end of the 1st year. For the majority of infants in the long-term MMF cluster, the first day of MMF was relatively early in life, recorded within the first month after birth. Most infants in the mostly BF cluster, in contrast, were introduced to MMF after (or sometimes before) at least a few weeks of exclusive BF. Considering the reduction in breast milk supply once FF is introduced, it would be relevant to understand which factors allowed or supported this continued BF such as physiological factors, intended BF duration, BF support, or specific MMF practices (e.g., alternating feeds or topping-up feeds and proportion of night vs. day feeds). However, as both the TEMPO and Venus studies were not designed to study MMF behaviour, these data on maternal BF intentions, reasons for FF introduction, type of BF support received, MMF practices, etc., have not been collected and, therefore, do not allow for such an analysis.

Higher levels of maternal education were associated with more BF in both studies. This suggests that by improving maternal education, alongside maternal support, including BF coordinator input, BF rates, and/or durations, could potentially be improved. The positive association with maternal education is also seen in other surveys in Europe [32], and in Singapore, maternal education status has been found to be one of the most important determinants of BF duration [33].

In contrast to these findings in Europe and Singapore, a recent review of data from 81 low- and middle-income countries found early BF initiation and exclusive BF rates to have increased over the past two decades across all education categories [6]. The increases were, however, more prominent for “early BF initiation” in women with no formal education background and for “exclusive BF” in women with higher education levels. In almost all of these low- and middle-income countries, the use of formula was higher among women with the highest education levels. This may be due to the associated higher income, enabling the mothers to afford FF, or, as suggested by the authors, the increased use of FF by these women may be attributed to the increased participation of women in the labour force due to the improvements in maternal education, whereby FF may be used by mothers who return to work. While continuing to BF when they are with their child, they likely use FF while at work [6]. Maternal education (and associated higher employment rates) should thus be combined with BF-supportive conditions and worksite support, such as adequate maternity leave, paid lactation breaks, flexible working hours, and BF facilities, in order to reduce BF barriers and facilitate continued BF [34].

Exclusive BF is universally recognised for its benefits to both infants and mothers and should undeniably be promoted and supported as the first and foremost feeding choice for all infants. At the same time, due to the widespread MMF reality demonstrated in the described studies, a deeper understanding of this trend and its potential impact on BF success and other short- and long-term outcomes is relevant. When mothers introduce MMF based on choice or (perceived) necessity, continued and prolonged BF can extend some of the benefits of BF to both the mother and child as compared to complete BF cessation or a transition to exclusive FF. Whaley et al. found that in employees of the US Women, Infant, Child (WIC) program, delayed introduction of FF was associated with greater continued BF success at 12 months [34]. The timing of introduction of FF in this case affected total BF duration. A Swedish study similarly found that the earlier that FF was introduced, the shorter the BF duration [35]. In another study in mothers who supplemented BF with FF, doing so after 4 weeks of exclusive BF was strongly associated with longer durations of any BF, than doing so before 4 weeks of exclusive BF [36]. In the current study of the populations investigated, we found that the majority of mothers who provided FF before 1 month postpartum went on to cease BF soon after (“early exclusive FF” group). However, as highlighted above, another substantial proportion of mothers who introduced FF before 1 month continued BF for 12 months or even longer (“long-term MMF” group). In the current study we could not reliably investigate the determinants of BF or BF duration success in the MMF context such as the timing of introduction of FF, maternal BF intentions and goals, or maternal BF and FF drivers. Yet, these remain important factors to take into account to provide evidence-based advice for effective BF and BF continuation promotion strategies. Furthermore, a better understanding of the relationship between feeding clusters and health outcomes is highly valuable for future infant feeding research. The proposed methodology may be considered as a possible starting point for comparing milk feeding patterns among different populations to inform the design of future studies regarding infant feeding and potentially to study the impact of different MMF patterns on health and other outcomes.

### Strengths and Limitations

The statistical approach used for identifying the MMF clusters had the advantage of being data driven (with clustering methods being referred to as part of “unsupervised” machine learning). The fact that each subject is assigned to a cluster according to their feeding pattern without an a priori specified rule for doing so is a strong point of the illustrated method. Descriptive statistics were used to discuss differences and similarities between the different subsets corresponding to the mixed feeding clusters. Due to the exploratory nature of this work, no formal statistical testing was carried out.

The population considered in this paper comprised subjects who participated in a randomised controlled trial. This may induce a selection effect, by which the feeding patterns hereby analysed may not be identical to what is observed in different contexts. Similarly, the study population was from developed areas or upper-middle- and high-income countries. Therefore, the extrapolation of the results to other populations should be conducted with care.

Compared to the VENUS study, the prevalence of exclusive BF was higher in the TEMPO study (21% vs. 8%). Potentially, mothers with an infant with an increased risk of allergy development, as recruited in the TEMPO study, might choose to BF more and/or longer than mothers of infants not at risk to protect their infants from developing an allergy. Although more data are needed to understand whether such a relationship between BF degree/duration and infant allergy risk exists and to what extent, there are indirect suggestions that mothers of infants with allergic risk BF more. Some researchers have speculated that women who are aware that their child has an increased risk of developing allergy may BF for longer [37]. Interpretation of the current results should therefore keep this potential bias in mind.

In both studies, the raw feeding data were compiled from parent reported questionnaires and diaries. Because the data collection took place outside a clinical setting, and at several time points, inaccuracies may be present even after data cleaning. Despite this, the current results are likely to be relatively robust if the overall feeding pattern during the first year of life was captured with reasonable correctness. For future studies, the use of more sophisticated tools, such as smart feeding bottles, to automatically and digitally record when and how much milk was given, would improve the accuracy of the recorded milk intakes, at least as far as bottle feeding formula and expressed breast milk is concerned.

## 5. Conclusions

In conclusion, by bringing structure to heterogeneous datasets we have described different current MMF practices and patterns. The clustering methodology showed reliable use in populations from both Europe and Southeast Asia. These insights have shed light on current trends and feeding practices in different regions and could lay the foundation for future studies investigating the relationship between different infant feeding practices and health and other outcomes. Our findings indicate a positive relationship between maternal education and BF, underscoring the importance of recognising this modifiable risk factor in relation to infant feeding practices. Finally, we stress the importance and relevance of a unified, agreed term to describe the combined feeding of breastmilk and infant formula. Such an institutionally agreed upon definition will allow for standardised comparisons across studies, enabling the build-up of the evidence base around the spectrum of different infant feeding practices and their effects.

## Figures and Tables

**Figure 1 nutrients-14-02190-f001:**
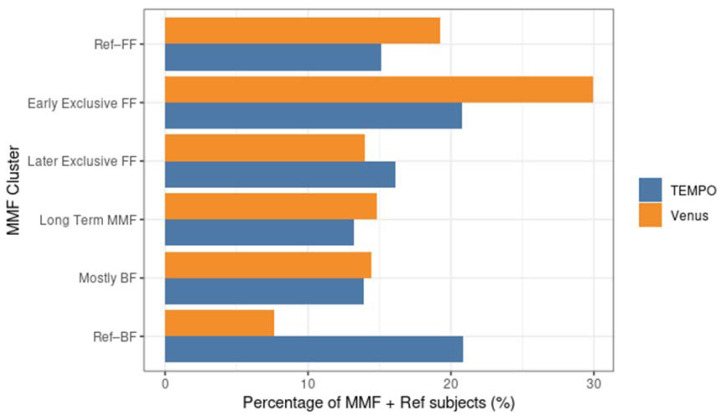
Percentage of subjects in different MMF clusters and reference groups in the MMF + Ref population for Tempo and Venus. FF: formula feeding; BF: breastfeeding; MMF: Mixed milk Feeding; Ref–BF: Subjects who had no, or limited infant milk formula intake recorded during the first year (on ≤5% of total recorded days).

**Figure 2 nutrients-14-02190-f002:**
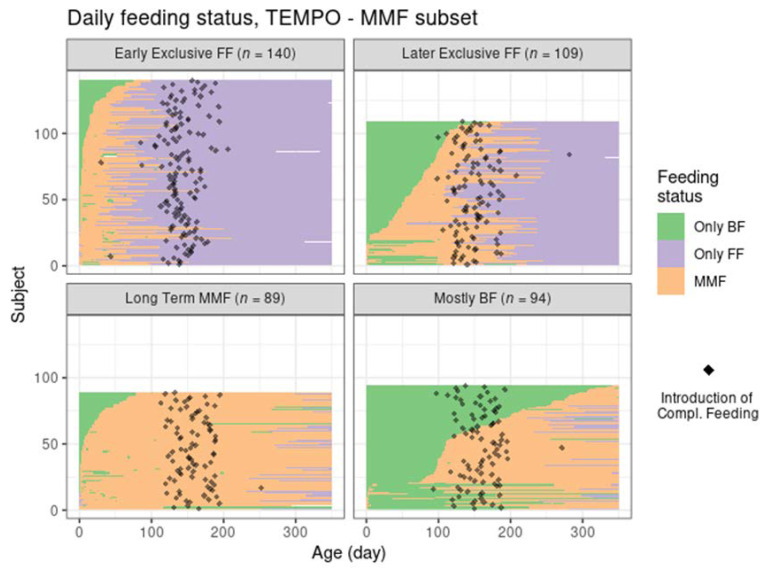
Feeding status trajectory for all MMF subjects in TEMPO per cluster. Each subject’s feeding pattern is represented by a horizontal line ranging from birth (leftmost) to 350 days of age (rightmost). The colour at each age describes whether the subject was exclusively BF, exclusively FF, or mixed feeding. The black dots indicate the recorded day of starting complementary feeding.

**Figure 3 nutrients-14-02190-f003:**
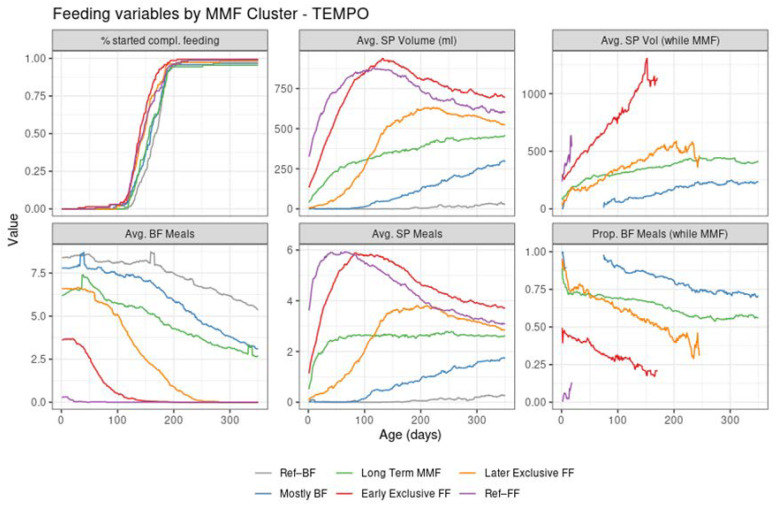
Summary of feeding variables in TEMPO per cluster from birth (leftmost) to 350 days of age (rightmost). First row: % started compl. feeding—percentage of subjects introduced to complementary feeding; Avg. SP Volume (mL)—average of the estimated daily study product volume intake across all subjects in the cluster; Avg. SP Vol (while MMF)—average of the estimated study product volume intake in ml per day, across the subjects in the cluster who are mixed feeding at that time point. Second row: Avg. BF Meals—average of the estimated daily number of daily BF meals; Avg. SP Meals—average of the estimated daily number of daily study product meals; Prop. BF Meals (while MMF)—proportion of BF meals from the total of BF meals and study product meals.

**Figure 4 nutrients-14-02190-f004:**
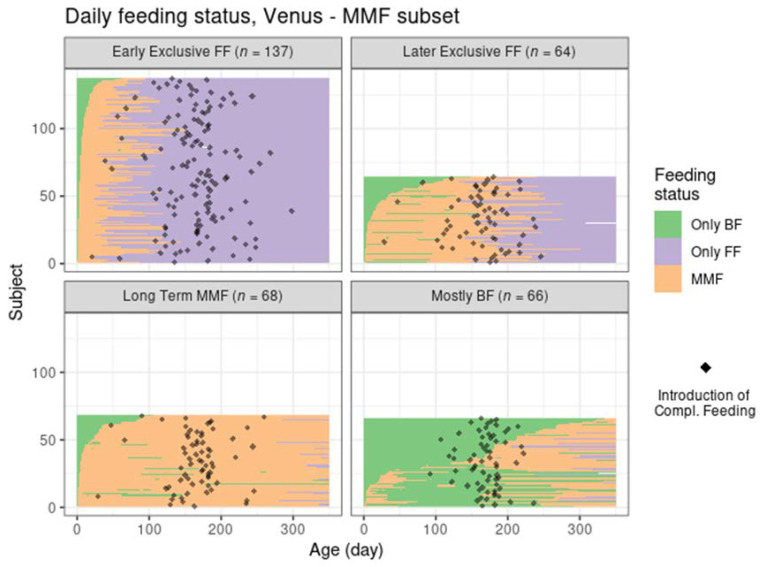
Feeding status trajectory for all MMF subjects in Venus per cluster. Each subject’s feeding pattern is represented by a horizontal line ranging from birth (leftmost) to 350 days of age (rightmost). The colour at each age describes whether the subject was exclusively BF, exclusively FF, or mixed feeding. The black dots indicate the recorded day of starting complementary feeding.

**Figure 5 nutrients-14-02190-f005:**
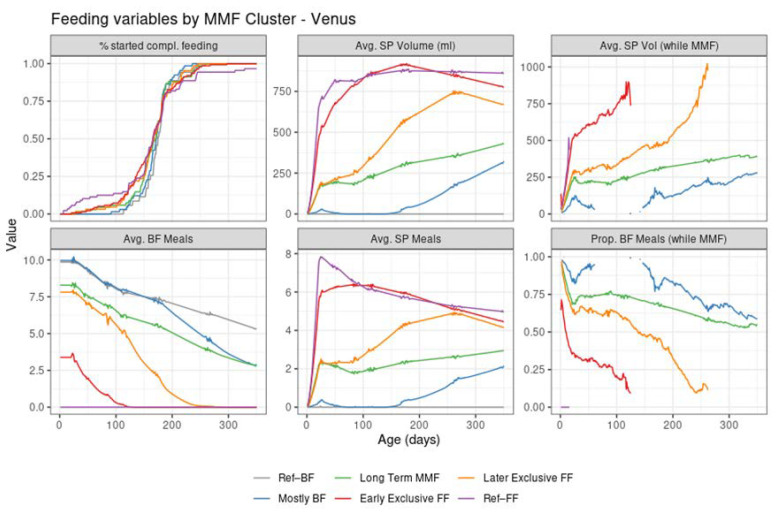
Summary of feeding variables in Venus per cluster from birth (leftmost) to 350 days of age (rightmost). First row: % started compl. Feeding—percentage of subjects that introduced complementary feeding; Avg. SP Volume (mL)—average of estimated daily study product volume intake across all subjects in the cluster; Avg. SP Vol (while MMF)—average of estimated study product volume intake in ml per day across the subjects in the cluster who are mixed feeding at that time point. Second row: Avg. BF Meals—average of the estimated daily number of daily BF meals; Avg. SP Meals—average of the estimated daily number of daily study product meals; Prop. BF Meals (while MMF)—proportion of BF meals from the total of BF meals and study product meals.

**Figure 6 nutrients-14-02190-f006:**
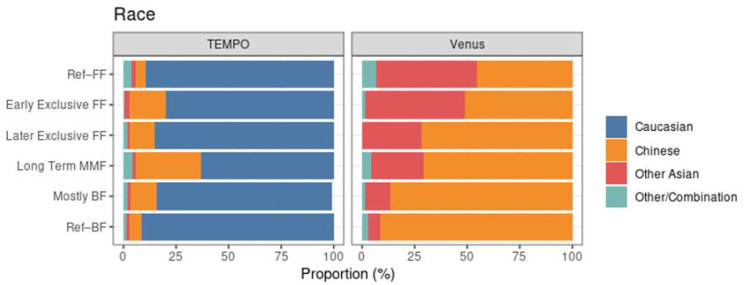
Race distribution within each MMF cluster, highlighting the population differences between TEMPO (left) and Venus (right).

**Figure 7 nutrients-14-02190-f007:**
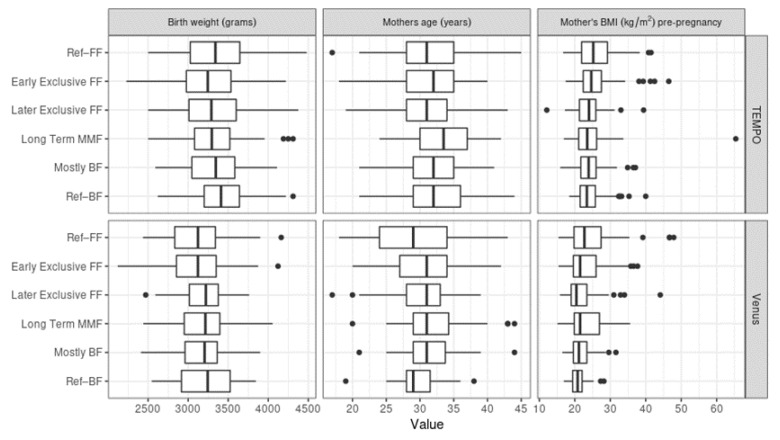
Similarities between birth weight, maternal age, and maternal body mass index (BMI) pre-pregnancy among the MMF Clusters. The three variables are shown for TEMPO (above) and Venus (below).

**Figure 8 nutrients-14-02190-f008:**
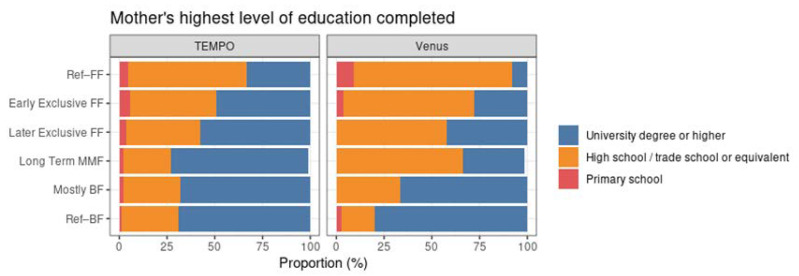
Distribution of mothers’ highest level of education completed within each MMF cluster for TEMPO (left) and Venus (right).

**Table 1 nutrients-14-02190-t001:** Group sizes in TEMPO, including the percentage out of all of the subjects enrolled.

Feeding Group	*n*	Percentage
Ref–BF	141	16.5%
Ref–FF	102	11.9%
MMF (>21 days MMF)	432	50.5%
Short transition (<21 days MMF)	92	10.8%
Follow up < 300 days	88	10.3%
Total	855	100%

Ref–BF: Subjects who had no, or limited infant milk formula intake recorded during the first year (on ≤5% of total recorded days); Ref–FF: Subjects who had no, or limited breast milk intake recorded during the first year (on ≤5% of total recorded days); MMF: Mixed milk Feeding; TEMPO: prospective, randomised, controlled, multicentre study that was conducted in 13 dif-ferent countries in Europe and Asia.

**Table 2 nutrients-14-02190-t002:** Group sizes in Venus, including the percentage out of all of the subjects enrolled.

Feeding Group	*n*	Percentage
Ref–BF	35	6.5%
Ref–FF	88	16.3%
MMF (>21 days MMF)	335	62.2%
Short transition (<21 days MMF)	30	5.6%
Follow up < 300 days	16	3%
No feeding records	35	6.5%
Total	539	100%

Venus: a randomised controlled trial carried out in a population of healthy Asian infants.

**Table 3 nutrients-14-02190-t003:** Size of the MMF clusters and reference groups in TEMPO and Venus, including the percentage out of the total MMF and reference group subjects.

MMF Cluster	*n* (TEMPO)	Percentage (TEMPO)	*n* (Venus)	Percentage (Venus)
Ref–BF	141	20.9%	35	7.6%
Mostly BF	94	13.9%	66	14.4%
Long-term MMF	89	13.2%	68	14.8%
Later exclusive FF	109	16.1%	64	14%
Early exclusive FF	140	20.7%	137	29.9%
Ref–FF	102	15.1%	88	19.2%
Total	675	100%	458	100%

## Data Availability

The data presented in this study are available upon request from the corresponding author. Requests will be accepted from qualified researchers that meet the Danone Nutricia Research criteria for access to confidential data (laid out in the Danone Nutricia Clinical Data Sharing policy).

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
