# Peer review of "Mixed Milk Feeding: A New Approach to Describe Feeding Patterns in the First Year of Life Based on Individual Participant Data from Two Randomised Controlled Trials"

_nutrients, 2022, doi:10.3390/nu14112190_

Round 1
Reviewer 1 Report
The authors address a relevant and complex issue. Limited information is available with respect to mixed feeding patterns in practice on an individual level. The work uses an interesting approach to handle the diversity of the data and start the discussion on a more detailed categorization in case of mixed feeding practice. This will eventually lead to more targeted interventions to increase breastfeeding practice and facilitate interpretation of studies with mixed feeding practice.
How did the authors classify the feeding practice in which breast milk is provide to the infant via bottle?
Author Response
- Comment 1: The authors address a relevant and complex issue. Limited information is available with respect to mixed feeding patterns in practice on an individual level. The work uses an interesting approach to handle the diversity of the data and start the discussion on a more detailed categorization in case of mixed feeding practice. This will eventually lead to more targeted interventions to increase breastfeeding practice and facilitate interpretation of studies with mixed feeding practice.
- Response 1: We’d like to thank the reviewer for their supportive comments emphasizing the relevance of our work.
- Comment 2: How did the authors classify the feeding practice in which breast milk is provide to the infant via bottle?
- Response 2: No distinction was made regarding the way in which the breast milk meals were provided to the infants.
Reviewer 2 Report
Review 1679118
Mixed Milk Feeding: a new approach to describe feeding patterns in the first year of life based on individual participant data from two Randomized Controlled Trials
Thank you for the opportunity to review this manuscript. The differences in infant and maternal health between exclusive breastfeeding and exclusive formula feeding are well-researched. However, as this manuscript demonstrates, many infants receive both breast milk and infant formula and the proportions of each are not usually quantitated. This manuscript seeks to fulfil a much-needed objective categorisation of different types of mixed milk feeding that can form the basis for research into the infant and maternal outcomes depending on what proportion of breast milk and formula is fed.
Statistics is not my strong point, but in the results section various differences were described as “higher”, “earlier” or “later” without any reference to the statistical test or probability values.
It is interesting to see “Study Product” (infant formula) volumes presented for the different MMF groups. The number of formula meals and breast milk meals is also presented, but a breast milk meal is not a standard quantity. The authors should acknowledge that the volume of breast milk taken by the infant has not been quantitated.
On page 12 the statements on lines 428-434 should be referenced.
Author Response
- We’d like to thank the reviewer for their supportive comments emphasizing the relevance of our work.
- Comment 1: Statistics is not my strong point, but in the results section various differences were described as “higher”, “earlier” or “later” without any reference to the statistical test or probability values.
- Response 1: We’d like to thank the reviewer for their comment. We aim to describe the observed results by using different descriptive methods, such as graphs and tables, included either in the main body of the manuscript or the supplementary material. This is why the differences (or similarities) observed between the different feeding clusters are not described as statistically significant.
- The choice of not referencing formal statistical tests and p-values is due to the exploratory rather than confirmatory nature of the work presented in this manuscript. In the absence of pre-specified hypotheses and effect sizes of interests, the p-values would be of limited usefulness, as outlined for example in the American Statistical Association’s statement on p-values (Ronald L. Wasserstein & Nicole A. Lazar (2016) The ASA Statement on p-Values: Context, Process, and Purpose, The American Statistician, 70:2, 129-133, DOI: 10.1080/00031305.2016.1154108).
- To clarify this approach, we have now added the following to the Discussion / Strengths & Limitations section:
- “Descriptive statistics were used to discuss differences and similarities between the different subsets corresponding to the mixed feeding clusters. Due to the exploratory nature of this work, no formal statistical testing was carried out.”
- Comment 2: It is interesting to see “Study Product” (infant formula) volumes presented for the different MMF groups. The number of formula meals and breast milk meals is also presented, but a breast milk meal is not a standard quantity. The authors should acknowledge that the volume of breast milk taken by the infant has not been quantitated.
- Response 2: Thank you, we have now clarified this in Section 2:
- “The average daily volume of study product (‘SP’) intake was also recorded on a weekly basis, together with other data such as daily number of study product feedings and daily number of breast milk feedings. However, a breast milk meal is not a standard quantity and the volume intake of breast milk (or of formula other than study product) were not recorded.”
- Comment 3: On page 12 the statements on lines 428-434 should be referenced.
- Thank you for pointing out this clear oversight. We have now added the 3 relevant references (#34-#36).
Reviewer 3 Report
The manuscripts described four types of mixed milk feeding(MMF) practice by using cluster analysis. The study subjects were from two randomized controlled trials (‘TEMPO’ and ‘Venus’). It is quite necessary to define MMF clearly for future studies and the study did a try.
- The defintion (lines 56-58) of MMF from World Health Organization could be updated with the newly released WHO infant and young child feeding indicators.
- After 6 months of age, complementary foods will be recommended. If formula would be considered as a complementary food after 6 months, definition of MMF may be not necessary then.
- The cutoff of 21 days was used for defining MMF and sensitivity analysis of 7, 14 and 28 days was conducted with similar cluster interpretation. It might be worthwile to test without any restriction of MMF.
- MMF cluster was data-driven. Are characteristics of each cluster same for ‘TEMPO’ and ‘Venus’? If not, comparision would be needed.
- “Figure 3. Summary of feeding variables in TEMPO. …… Avg. SP Volume (ml) - average of estimated daily study product volume intake; Avg. SP Vol (while MMF) - average of estimated study product volume intake in ml per 277 day, only for subjects that are mixed feeding”. SP Volume (ml) would be similar as Avg. SP Vol (while MMF) for those MMF clusters.
- Figure 4 is identical as figure 2 in terms of sample size. Figure 5 is same as figure 3. Are there some errors.
- Are there some statistical analyses in figure 6 and figure 7? The two studies were quite difference in terms of study populations. It may not be valid to show the description difference without further analyses.
- It would be critical to focus on the description of MMF rather than to explore other potential relationship in conclusiton.
- The samples for ‘Venus’ in the abstract (n=855) were not same as the one in the main text (n=458).
Author Response
We’d like to thank the Reviewer for their constructive comments and suggestions.
Please see below our response to the Reviewer's comments, as requested:
- Comment 1: The definition (lines 56-58) of MMF from World Health Organization could be updated with the newly released WHO infant and young child feeding indicators.
Response 1: Thank you for pointing out these updated definitions. We have now improved the manuscript with the most recent Mixed Milk Feeding WHO definition (Lines 56 – 59) and added the reference above to the manuscript.
- Comment 2: After 6 months of age, complementary foods will be recommended. If formula would be considered as a complementary food after 6 months, definition of MMF may be not necessary then.
Response 2: Formula was not considered as complementary food in the two studies.
We have now clarified this in Section 2, where we added the sentence “Note that infant formula or follow-up formula were not considered complementary foods.”
- Comment 3: The cutoff of 21 days was used for defining MMF and sensitivity analysis of 7, 14 and 28 days was conducted with similar cluster interpretation. It might be worthwile to test without any restriction of MMF.
Response 3: Thank you for your constructive comment. This is indeed one of the approaches that were also considered at earlier points in the research. We have examined applying the clustering approach to all the subjects, without any restrictions to MMF (equivalent to 0 days definition). The 4 feeding patterns identified in this way were also very similar to the main results in the paper, with a few differences (such as no more Ref – BF and Ref – FF groups, subjects which with this other approach are also assigned to one of the 4 clusters). In the end, we felt that applying the clustering algorithm to the mixed feeding subset is more relevant for the purposes of this paper.
We have now clarified this in the text in Section 4 (Discussion):
- “As a sensitivity analysis, other cut-offs were also employed using 0, 7, 14 and 28 days and they resulted in clusters that generally follow the same interpretation as with the 21 day definition. Note that in the special case of 0 days, all subjects are assigned to a cluster, in which case the clusters do not refer to the MMF subset, but rather the whole population in the two studies.”
- Comment 4: MMF cluster was data-driven. Are characteristics of each cluster same for ‘TEMPO’ and ‘Venus’? If not, comparision would be needed.
Response 4: Thank you for considering this point. Indeed, the characteristics of the MMF clusters in TEMPO and Venus are generally similar with regards to the observed feeding patterns and the distribution of baseline variables, despite the different settings of the two clinical trials.
We have described the similarities and differences in terms of feeding patterns in Section 3.2 of the paper, and more details are provided in Figures S5, S6, and Table S1 in the Supplementary Material.
The similarities and differences in terms of baseline variables are discussed in Section 3.3 of the paper, with detailed comparisons shown in Tables S3 and S5 in the supplementary material.
Comment 5: “Figure 3. Summary of feeding variables in TEMPO. …… Avg. SP Volume (ml) - average of estimated daily study product volume intake; Avg. SP Vol (while MMF) - average of estimated study product volume intake in ml per 277 day, only for subjects that are mixed feeding”. SP Volume (ml) would be similar as Avg. SP Vol (while MMF) for those MMF clusters.
Response 5: The two variables are defined as follows:
- Avg. SP Volume (ml) is the average study product intake over all the subjects in a cluster at a given age.
- Avg. SP Vol (while MMF) is the average study product intake over the subjects who are mixed feeding in a cluster at a given age.
The two quantities are equal at a given age in a cluster only if all subjects are mixed feeding at that given age.
We have now clarified this in the captions of Figure 3 and Figure 5 as follows:
“Avg. SP Volume (ml) - average of estimated daily study product volume intake across all subjects in the cluster; Avg. SP Vol (while MMF) - average of estimated study product volume intake in ml per day, across the subjects in the cluster who are mixed feeding at that time point.”
- Comment 6: Figure 4 is identical as figure 2 in terms of sample size. Figure 5 is same as figure 3. Are there some errors.
Response 6: We apologize for our editing mistake This was an unfortunate drafting error in the manuscript. Please find the corrected graphs in the updated version of the manuscript.
- Comment 7: Are there some statistical analyses in figure 6 and figure 7? The two studies were quite difference in terms of study populations. It may not be valid to show the description difference without further analyses.
Response 7: Thank you for this comment. Figures 6 and 7 show descriptive plots to illustrate similarities and differences between the MMF clusters and reference groups in the two studies. These variables were selected on the basis of their perceived relevance for the topic of mixed milk feeding. We aimed to describe the observed differences by using descriptive methods, such as graphs and tables, included either in the main body of the manuscript or the supplementary material. This is why the differences (or similarities) observed between the different feeding clusters are not described as statistically significant.
The choice of not referencing formal statistical tests and p-values is due to the exploratory rather than confirmatory nature of the work presented in this manuscript. In the absence of pre-specified hypotheses and effect sizes of interests, the p-values would be of limited usefulness, as outlined for example in the American Statistical Association’s statement on p-values (Ronald L. Wasserstein & Nicole A. Lazar (2016) The ASA Statement on p-Values: Context, Process, and Purpose, The American Statistician, 70:2, 129-133, DOI: 10.1080/00031305.2016.1154108).
To clarify this we have now added the following in the Discussion / Strengths & Limitations section:
- “Descriptive statistics were used to discuss differences and similarities between the different subsets corresponding to the mixed feeding clusters. Due to the exploratory nature of this work, no formal statistical testing was carried out.”
- Comment 8: It would be critical to focus on the description of MMF rather than to explore other potential relationship in conclusiton.
Response 8: We appreciate the comment. At the same time, it is our conviction that these clusters can lay the foundation for studying associations between feeding patterns and (health) outcomes. We have now softened this point by rewording the Conclusion as follows:
- “In conclusion, by bringing structure to heterogeneous datasets we have described different current MMF practices and patterns. The clustering methodology showed reliable use in various geographies. These insights have shed light on current trends and feeding practices in different regions, and could lay the foundation for future studies investigating the relationship between different infant feeding practices and health and other outcomes.”
- Comment 9: The samples for ‘Venus’ in the abstract (n=855) were not same as the one in the main text (n=458).
Response 9: We apologize for this error and have now corrected it in the manuscript. The population numbers in the abstract now refer to the ASE (all subject enrolled) population in the two studies; Tempo n = 855 and Venus n = 539. The MMF + BF Reference groups comprise 675 in Tempo and 458 in Venus (as shown in Table 3).
Reviewer 4 Report
There are problems with the submission that have created a fatal flaw. Essentially Figures 4 and 5 which are supposed to be for the VENUS study are actually copies of Figures 2 and 3. Thus, the paper and its findings cannot be reviewed.
The pattern labelled "late exclusive FF" is misleading -- "later" could be ok.
There are incorrect statements in the para starting on line 85. The authors do not recognize that there is a global recommendation regarding exclusive breastfeeding for 6 months and that the evidence which supports that global recommendation comes from comparisons of exclusive breastfeeding with other patterns, including MMF, and exclusive FF. Therefore, the statement that the these patterns are not easy understood and that we don't know what happens regarding the benefits of MMF or of BF may be not correct.
Author Response
We’d like to thank the Reviewer for their constructive comments and suggestions.
Please see below our response to the Reviewer's comments, as requested:
- Comment 1: There are problems with the submission that have created a fatal flaw. Essentially Figures 4 and 5 which are supposed to be for the VENUS study are actually copies of Figures 2 and 3. Thus, the paper and its findings cannot be reviewed.
Response 1: We apologize for our editing mistake. Please find the corrected graphs in the updated version of the manuscript. The graphs submitted separately together with the paper were correct, but due to a copy paste error the same figures were pasted twice in the manuscript.
- Comment 2: The pattern labelled "late exclusive FF" is misleading -- "later" could be ok.
Response 2: Thank you for your suggestion of improvement. We agree that “later” reflects the context better than “late” and have made the relevant adjustments to the graphs and text.
- Comment 3: There are incorrect statements in the para starting on line 85. The authors do not recognize that there is a global recommendation regarding exclusive breastfeeding for 6 months and that the evidence which supports that global recommendation comes from comparisons of exclusive breastfeeding with other patterns, including MMF, and exclusive FF. Therefore, the statement that the these patterns are poorly understood and that we don't know what happens regarding the benefits of MMF or of BF is incorrect and misleading.
Response 3: Thank you for pointing out some unclarities and omissions in the paragraph starting in Line 85. We have now included the statement that the WHO recommends 6 months of exclusive breastfeeding. Furthermore we have enhanced clarification that, while the effects of non-exclusive breastfeeding (grouped as a whole) have been well documented in comparison to exclusive breastfeeding, the contribution/effect of mixed milk feeding (only with infant formula) as a specific subgroup, has not been well studied. This has been adjusted in the relevant paragraph in the Introduction.
Round 2
Reviewer 3 Report
The authors addressed the comments raised in the previous review well. One concern is that the study population were from developed areas or middle income countries. The situation may not be similar in developing world. This needs to be addressed in the limitation and conclusion section.
Author Response
We'd wish to thank the Reviewer for raising this very valid point about results not automatically being generalisable to the developing world.
We've now adjusted the manuscript in Lines 480-481 as well as Line 509 as follows:
"The population considered in this paper comprised subjects who participated in a randomized controlled trial. This may induce a selection effect, by which the feeding patterns hereby analysed may not be identical to what is observed in different contexts. Similarly, the study population was from developed areas or upper-middle and high income countries. Therefore, the extrapolation of the results to other populations should be done with care."
*******************
The clustering methodology showed reliable use in populations from both Europe and Southeast Asia (removed "various geographies").